# Comparative Genomics Analysis Provides New Insights into High Ethanol Tolerance of *Lactiplantibacillus pentosus* LTJ12, a Novel Strain Isolated from Chinese Baijiu

**DOI:** 10.3390/foods12010035

**Published:** 2022-12-22

**Authors:** Jiali Wang, Chengshun Lu, Qiang Xu, Zhongyuan Li, Yajian Song, Sa Zhou, Le Guo, Tongcun Zhang, Xuegang Luo

**Affiliations:** 1Key Laboratory of Industrial Fermentation Microbiology of the Ministry of Education, College of Biotechnology, Tianjin University of Science and Technology, Tianjin 300457, China; 2Ningxia Key Laboratory of Clinical and Pathogenic Microbiology, General Hospital of Ningxia Medical University, Yinchuan 750004, China

**Keywords:** *Lactiplantibacillus pentosus* LTJ12, whole genome sequencing, gene function annotation, probiotics, ethanol stress, comparative genomics

## Abstract

Lactic acid bacteria have received a significant amount of attention due to their probiotic characteristics. The species *Lactiplantibacillus plantarum* and *Lactiplantibacillus pentosus* are genotypically closely related, and their phenotypes are so similar that they are easily confused and mistaken. In the previous study, an ethanol-resistant strain, LTJ12, isolated from the fermented grains of soy sauce aroma type baijiu in North China, was originally identified as *L. plantarum* through a 16S rRNA sequence analysis. Here, the genome of strain LTJ12 was further sequenced using PacBio and Illumina sequencing technology to obtain a better understanding of the metabolic pathway underlying its resistance to ethanol stress. The results showed that the genome of strain LTJ12 was composed of one circular chromosome and three circular plasmids. The genome size is 3,512,307 bp with a GC content of 46.37%, and the number of predicted coding genes is 3248. Moreover, by comparing the coding genes with the GO (Gene Ontology), COG (Cluster of Orthologous Groups) and KEGG (Kyoto Encyclopedia of Genes and Genomes) databases, the functional annotation of the genome and an assessment of the metabolic pathways were performed, with the results showing that strain LTJ12 has multiple genes that may be related to alcohol metabolism and probiotic-related genes. Antibiotic resistance gene analysis showed that there were few potential safety hazards. Further, after conducting the comparative genomics analysis, it was found that strain LTJ12 is *L. pentosus* but not *L. plantarum*, but it has more functional genes than other *L. pentosus* strains that are mainly related to carbohydrate transport and metabolism, transcription, replication, recombination and repair, signal transduction mechanisms, defense mechanisms and cell wall/membrane/envelope biogenesis. These unique functional genes, such as gene 2754 (encodes alcohol dehydrogenase), gene 3093 (encodes gamma-D-glutamyl-meso-diaminopimelate peptidase) and some others may enhance the ethanol tolerance and alcohol metabolism of the strain. Taken together, *L. pentosus* LTJ12 might be a potentially safe probiotic with a high ethanol tolerance and alcohol metabolism. The findings of this study will also shed light on the accurate identification and rational application of the *Lactiplantibacillus* species.

## 1. Introduction

Probiotics are microbial supplements that have a variety of physiological effects on the body, improving the balance of intestinal microorganisms [1,2]. *Lactobacillus* are widely employed as probiotics in the food industry [3]. Later, in the 20th century, the classification of new species was mainly carried out based on genotypic and chemotaxonomic criteria. The 16S rRNA gene sequence similarity analysis was always used as the classification basis for *Lactobacillus*. Recently, with in-depth research, people found that 16S rRNA analysis could not distinguish the phylogenetic relationship between different cladistic lactic acid bacteria; therefore, some of the *Lactobacillus* were reclassified and changed to *Lactiplantibacillus*, which are Gram-positive, non-spore-forming, homofermentative and non-motile rods. *Lactiplantibacillus* can ferment a wide range of carbohydrates, and most species of this new genus can metabolize phenolic acids by esterase, decarboxylase and reductase activities. *Lactiplantibacillus* include 16 species, including *Lactiplantibacillus plantarum*, *Lactiplantibacillus pentosus*, *Lactiplantibacillus paraplantarum*, etc. However, *L. plantarum* and *L. pentosus* are very similar in terms of the gene and functional phenotype, which is very easy to be confused and mistaken. There are differences in the *recA* genes between the two species, so it can be used as one of the methods to distinguish them. *L. pentosus* is an important member of LAB and is widely distributed in traditional fermented foods and the gut. Due to their health-promoting effects, more and more strains of *L. pentosus* are widely used as probiotics in food and dietary supplements [4].

With the rapid development of science and technology, researchers exploring and studying LAB have also begun to explore the basic research of some physiological and biochemical experiments in the field of related mechanism research involving the research of genomics, transcriptomics and metabolomics [5,6]. Through sequence analysis and its functional annotation, it is possible to better understand the variation within and between species [7]. Comparative genomics is a field of biological research in which researchers use a variety of tools to compare the complete genome sequences of different species. By carefully comparing characteristics that define various organisms, researchers can pinpoint regions of similarity and difference (https://www.genome.gov/about-genomics/fact-sheets/Comparative-Genomics-Fact-Sheet, accessed on 1 July 2022). Liu et al. [8] sequenced the genome of *L. plantarum* 5-2, which was derived from fermented soybean isolated from Yunnan province, China. The genome encodes the key enzymes required for the EMP and phosphoketolase pathways, and an extracellular serine protease existed in 5-2 different with *L. plantarum* WCSF1. Furthermore, Shi et al. [9] selected two strains isolated from fermented foods for characterisation of whole genome sequences. It was found that the *cpsD* gene cluster in *L. plantarum* CGMCC12436 may be closely related to its colonization ability in the mouse gut during and after discontinuing the strain administration. Additionally, Schmid et al. [7] used Pacific Biosciences’ long read technology to sequence and de novo assemble the genomes of three *Lactobacillus helveticus* starter strains. A genome mining effort uncovered examples in which an experimentally observed phenotype could be linked to the underlying genotype. Genomics is also particularly important in the classification of species. Members of the *L. plantarum* phylogenetic group, which consist of *L. plantarum*, *L. pentosus* and *L. paraplantarum*, are indistinguishable from their 16S rRNA gene sequences due to their 99% similarity [10]. Compared to the limitation of 16S rRNA gene sequences, the genome contains all the genetic information of the species. Therefore, through the analysis of genomics, they can be well separated, and the differences in their genomes can be found. Taxonomic and functional characterization of new isolates can be performed more accurately.

Currently, genome sequencing has been carried out to reveal the characteristics, physiological functions and metabolic mechanism of *L. pentosus* at the gene level. However, more functions, properties and future prospects of *L. pentosus* are not yet clear, and further in-depth research is needed. Ye et al. [11] analyzed the complete genome sequence of *L. pentosus* ZFM94 to highlight the probiotic features at a genetic level. Stergiou et al. [12] performed a comprehensive bioinformatic analysis and whole-genome annotation of a potential probiotic strain *L. pentosus* L33 isolated from fermented sausages, highlighting the genetic loci and probiotic phenotypes involved in host-microbe interactions. Ethanol is an organic solvent that can alter the permeability of cell membranes, cause the loss of intracellular molecules, and impair enzyme performance. Excessive ethanol concentration will change the physiological activity and metabolism of bacterial cells to varied degrees during the growth phase [13,14]. In the process of production and fermentation, LAB will be subject to a certain degree of environmental stress, which makes the required good LAB need not only good fermentation performance but also a strong tolerance [2]. The ability of LAB to withstand environmental stress, particularly high ethanol concentrations, is very promising. It is of great significance for the study of the mechanism of LAB against ethanol stress and alcohol metabolism.

*L. pentosus* LTJ12 was isolated from the fermented grains of soy sauce aroma type baijiu in North China. In our previous studies, through 16S rRNA sequence analysis, LTJ12 was identified as *L. plantarum* which had a high resistance to ethanol stress [15]. However, here, through whole-genome sequencing, we found that LTJ12 should be reclassified as *L. pentosus*. In order to further study the functional mechanism of anti-ethanol stress and the alcohol metabolism of *L. pentosus* LTJ12, the basic genome functions of the GO (Gene Ontology), COG (Cluster of Orthologous Groups) and KEGG (Kyoto Encyclopedia of Genes and Genomes) databases were annotated, so as to provide bioinformatics basis for further understanding the metabolic mechanism of LTJ12 under ethanol stress. In addition, we also performed a genetic analysis of the probiotic potential of the LTJ12 strain, and the LTJ12 strain was compared with other publicly available strain genomes.

## 2. Materials and Methods

### 2.1. Bacterial Strains

*L. pentosus* LTJ12 was separated and screened from the fermented grains of soy sauce aroma type baijiu in North China by our team. It was preserved in the Key Laboratory of Industrial Fermentation Microbiology of the Ministry of Education, College of Biotechnology, Tianjin University of Science and Technology.

### 2.2. Determination of Ethanol Degradation Ability and the Effect of Ethanol on the Growth of Strains

The strains were inoculated into MRS liquid medium containing different volume fractions (4%, 6%, 8%, 10%, 12%) of ethanol and cultured at 37 °C for 12 h. For the MRS ethanol liquid medium without strains as a control, the residual ethanol was determined using the potassium dichromate-sulfuric acid method. All experiments were performed three times. The difference between the initial ethanol and the residual ethanol was the content of degraded ethanol. The more the ethanol was degraded, the stronger the ethanol degradation ability was. Therefore, the ethanol degradation ability of each strain was analyzed.

The strains were cultured according to the above conditions, and the OD_600 nm_ value was measured every 2 h until 14 h to analyze the effect of ethanol on the growth of strains.

### 2.3. Extraction of Genomic DNA of L. pentosus LTJ12

The strain *L. pentosus* LTJ12 was cultured in a sterile MRS liquid medium at 37 °C for 12 h. Genomic DNA was extracted using Wizard^®^ Genomic DNA Purification Kit (Promega, Madison, WI, USA) according to the manufacturer’s protocol. Purified genomic DNA was quantified by TBS-380 fluorometer (Turner BioSystems Inc., Sunnyvale, CA, USA). High-quality DNA (OD260/280 = 1.8~2.0, >20 µg) was used to do further research.

### 2.4. Library Construction and Sequencing

Genomic DNA was sequenced using a combination of PacBio RS II Single Molecule Real Time (SMRT) and Illumina sequencing platforms. DNA samples were sheared into 400~500 bp fragments using a Covaris M220 Focused Acoustic Shearer following the manufacturer’s protocol. Illumina sequencing libraries were prepared from the sheared fragments using the NEXTflex^TM^ Rapid DNA-Seq Kit. Briefly, 5′ prime ends were first end-repaired and phosphorylated. Next, the 3′ ends were A-tailed and ligated to sequencing adapters. The third step was to enrich the adapters-ligated products using PCR. The prepared libraries were used for paired-end Illumina sequencing (2 × 150 bp) on an Illumina HiSeq X Ten machine. DNA fragments were then purified, end-repaired and ligated with SMRTbell sequencing adapters following the manufacturer’s recommendations (Pacific Biosciences, Menlo Park, CA, USA). The resulting sequencing library was purified three times using 0.45X volumes of Agencourt AMPure XP beads (Beckman Coulter Genomics, Danvers, MA, USA) following the manufacturer’s recommendations. Next, a~10 kb insert library was prepared and sequenced on one SMRT cell using standard methods.

### 2.5. Genome Assembly, Gene Prediction and Annotation

The data generated from the PacBio and Illumina platform were used for bioinformatics analysis. All of the analyses were performed using the free online platform of Majorbio Cloud Platform (www.majorbio.com, accessed on 1 June 2022) from Shanghai Majorbio Bio-pharm Technology Co., Ltd. (Shanghai, China).

The complete genome sequence was assembled using both the PacBio reads and Illumina reads. The original image data was transferred into sequence data via base calling, which is defined as raw data or raw reads and saved as a FASTQ file. A statistic of quality information was applied for quality trimming, by which the low-quality data can be removed to form clean data. The reads were then assembled into a contig using the hierarchical genome assembly process (HGAP) and canu [16]. The last circular step was checked and finished manually, generating a complete genome with seamless chromosomes and plasmids. Finally, error correction of the PacBio assembly results was performed using the Illumina reads with Pilon.

Glimmer [17] was used for CDS prediction, tRNA-scan-SE [18] was used for tRNA prediction, Barrnap was used for rRNA prediction, Infernal software (http://eddylab.org/infernal/, accessed on 1 June 2022) and Rfam database (https://rfam.xfam.org/, accessed on 1 June 2022) was used for sRNA prediction. The predicted CDSs were annotated from the NR, Swiss-Prot, Pfam, GO, COG and KEGG databases using sequence alignment tools such as BLAST, Diamond and HMMER. Briefly, each set of query proteins were aligned with the databases and annotations of the best-matched subjects (*e*-value < 10^−5^) were obtained for gene annotation. Carbohydrate utilization genes were annotated using the Carbohydrate Active Enzyme Database (CAZy) [19]. PathogenFinder (https://cge.cbs.dtu.dk//services/PathogenFinder/, accessed on 1 June 2022) [20] and CARD (Comprehensive Antibiotic Resistance Database, http://arpcard.Mcmaster.ca, Version 1.1.3, accessed on 1 June 2022) predicted disease-related genes and drug resistance-related genes [19].

### 2.6. Accession Number

All obtained raw sequence datasets have been uploaded to the NCBI Sequence Read Archive (SRA) with the accession number PRJNA907194.

### 2.7. Comparative Genomics Analysis

The genomes of 8 strains of *L. plantarum* and 8 strains of *L. pentosus* were selected respectively from NCBI for comparative genome analysis with strain LTJ12. The relationship between species was assessed at the genome-wide level by average amino acid identity (AAI). Pan-genome is the general term for all genes of a species. Through pan-genomics, the bacterial genome is analyzed from the perspective of the population, and the dynamic characteristics of bacterial genome are studied, so as to analyze the dynamic changes of bacterial genome in the process of evolution. All the above data were analyzed on the online tool of Majorbio Cloud Platform (https://cloud.majorbio.com/page/tools/, accessed on 1 July 2022) [21]. In addition, the genome of *L. pentosus* and *L. plantarum* WCFS1 were selected for collinearity analysis with LTJ12 in our study to check the multi-collinear relationship between the genomes and whether there is a multi-phenomenon of structural variation. The collinearity analysis of the strains was performed using Mauve software chimeric with the BLAST program operating on a JAVA platform [22].

### 2.8. Quantitative Real-Time PCR

For quantitative Real-time PCR, strain LTJ12 cultured respectively in MRS liquid medium with no ethanol and medium containing 8% ethanol for 12 h were collected. Total RNA was extracted by Trizol (Sigma, St. Louis, MO, USA) method. RNA was reverse transcribed into cDNA under the action of M-MLV reverse transcriptase (Promega, Madison, WI, USA). Reactions were performed on ABI-Step One^TM^ real-time PCR instrument (ABI, Los Angeles, CA, USA) with the SYBR Green qPCR Mastermix (DBI^®^ Bioscience, GER, Newark, DE, USA). PCR was performed under following conditions: 2 min initial denaturation at 95 °C, 40 cycles of 10 s denaturation at 95 °C, 60 °C anneal for 30 s, and 1 min extension at 95 °C, 55 °C for 1 min, finally 15 s extension at 95 °C. The primers used for qRT-PCR in our study were performed in Table 1. All experiments were performed three times. The 2^−ΔΔCT^ method was used for calculating the relative gene expression levels. 16S rRNA gene of *L. pentosus* was used as the internal control.

## 3. Results and Discussion

### 3.1. LTJ12 Has the Ability to Resist and Degrade Ethanol

In our previous study, the strain LTJ12 with high ethanol stress resistance were isolated from soy sauce aroma type baijiu in North China (Lutaichun, Tianjin) fermented grains. Meanwhile, LTJ30 is a strain that grows relatively poorly under ethanol stress. Through 16S rRNA sequence analysis, both LTJ12 and LTJ30 were identified as *L. plantarum* with high similarity [15]. Based on previous studies, the growth and ability to degrade ethanol of these two strains at different ethanol concentrations were further investigated (Figure 1). It was found that the growth of LTJ12 was better than that of LTJ30 at higher ethanol concentrations. At the same time, the two strains had a certain ability to degrade ethanol under different concentrations of ethanol stress. Thus, LTJ12 was a potential probiotic strain with both ethanol tolerance and ethanol degradation ability. Therefore, we performed a genome-level analysis of LTJ12 to gain a deeper understanding of its functional mechanism.

### 3.2. Reclassification of LTJ12 via Genomic Analysis

In our previous study, based on the 16S rRNA sequence analysis, the LTJ12 strain was identified as *L. plantarum* [15]. However, *L. plantarum*, *L. pentosus* and *L. paraplantarum* are 99% similar by 16S rRNA gene sequences, making them indistinguishable, and genomics is particularly important in species classification. We conducted an in-depth study of its genome. By comparing with databases based on house-keeping genes (*frr*, *infC*, *nusA*, *pgk*, *pyrG*, *rplA*, *rplB*, *rplC*, *rplE*, *rplF*, *rplK*, *rplL*, *rplM*, *rplN*, *rplP*, *rplS*, *rplT*, *rpoB*, *rpsB*, *rpsC*, *rpsE*, *rpsI*, *rpsJ*, *rpsK*, *rpsM*, *rpsS*, *smpB*, *tsf*), we found that LTJ12 had a higher homology with *L. pentosus* (Figure 2), and the subsequent AAI analysis also yielded the same results (detailed analysis was shown in 3.8). Therefore, the LTJ12 strain was reclassified as *L. pentosus*.

### 3.3. Genome Sequencing and Assembly of L. pentosus LTJ12

The genome analysis contributes to a clearer understanding of the functional mechanisms of bacterial resistance to ethanol stress and its alcohol metabolism. Therefore, we analyzed the genome of *L. pentosus* LTJ12 to decipher the genetic code involved in resistance to ethanol stress and its alcohol metabolism. The genome of *L. pentosus* LTJ12 was composed of one circular chromosome and three circular plasmids (Figure 3). The genome size was 3,512,307 bp with a GC content of 46.37%, and contained 3248 coding genes with a total length of 2,835,894 bp. The ratio of gene/genome was 80.74%. The intergenetic region length was 676,413 bp and the ratio of intergenetic length/genome was 19.26%, and the GC content in gene region and intergenetic region was 47.59% and 41.24%, respectively. In addition, the number of predicted tRNA, rRNA and sRNA were 73, 16 and 38, respectively. The predicted coding genes were functionally annotated by aligning them with 6 major databases (NR, Swiss-Prot, Pfam, COG, GO and KEGG). Gene annotation is mainly based on protein sequence alignment. The gene sequences were compared with various databases to obtain the corresponding functional annotation information. Among them, 2437 genes were annotated in the GO database, 1541 functional genes were annotated in the KEGG database and 2593 genes were annotated in the protein database COG (Table 2).

### 3.4. Functional Annotation of L. pentosus LTJ12

The number of COG categories was 4, the number of COG types was 19, and a total of 2593 genes in the coding genome were annotated in the COG database, accounting for 79.83% of the annotated genes. COGs were classified into 4 categories: (1) metabolism, (2) cellular processes and signaling, (3) information storage and processing, and (4) poorly characterized. A total of 926 genes related to metabolism were divided into 8 types. There were 412 genes related to cellular processes and signaling, which were divided into 7 types. There were 521 genes related to information storage and processing, which were divided into 3 types. However, there were also 750 poorly characterized genes, which were classified as 1 type. Metabolism COG clustering mainly focused on carbohydrate transport and metabolism (248 related genes), amino acid transport and metabolism (212 related genes), inorganic ion transport and metabolism (141 related genes), energy production and conversion (109 related genes) (Figure 4A). Among the energy production and conversion, gene0084, gene0110, gene0116, gene0184, gene1750, gene1771, gene2153, gene2767, gene2746, gene2754, gene2747 and gene2866 were predicted to encode genes of 12 different alcohol dehydrogenases that have an impact on the metabolism of alcohol. The diversity of functional annotations suggested the possibility that *L. pentosus* LTJ12 has the ability to metabolize ethanol.

According to the alignment results, a GO annotation was performed on LTJ12 in the database. The LTJ12 genome contained 3 major types: biological process (BP), cellular component (CC) and molecular function (MF). A total of 2437 genes were functionally annotated in the GO classification, accounting for 75.03% of all coding genes (3248). Among them, the number of genes related to the biological process was 1676; the number of genes related to the cellular component was 1249; and the number of genes related to the molecular function was 1900 (Figure 4B). At the level of biological process, 470 functional annotations were obtained, such as oxidation-reduction process (GO:0055114), regulation of transcription, DNA-templated (GO:0006355) and transmembrane transport (GO:0055085), etc. Among them, the number of genes related to the oxidation-reduction process was the largest with 202 genes. One hundred and ninety-three genes related to regulation of transcription, DNA-templated, and 110 genes related to transmembrane transport. In the biological process, one gene was related to the alcohol metabolic process (GO:0006066), which was bifunctional acetaldehyde-CoA/alcohol dehydrogenase (gene3232). Fifty-three functional annotations were obtained at the level of cellular component, among which the number of genes related to the integral component of the membrane (GO:0016021) was the largest. At the molecular function level, a total of 748 functional annotations were obtained, of which the 324 genes related to ATP binding (GO:0005524) were the most, followed by 284 genes related to DNA binding (GO:0003677). In the molecular function, there were six genes related to alcohol dehydrogenase (NAD) activity (GO:0004022), including acetaldehyde dehydrogenase (gene0318), alcohol dehydrogenase (gene1515, gene1669 and gene2866), oxidoreductase (gene2330), and bifunctional acetaldehyde-CoA/alcohol dehydrogenase (gene3232), two genes related to phosphotransferase activity, alcohol group as an acceptor (GO:0016773), including uridine kinase (gene1425) and phosphomevalonate kinase (gene1584), and one gene related to aryl-alcohol dehydrogenase (NAD+) activity (GO:0018456), including aryl-alcohol dehydrogenase (gene2747).

A total of 1541 genes of LTJ12 in the KEGG database were functionally annotated in 38 pathways of metabolism, cellular processes, genetic information processing, human diseases, environmental information processing and organismal systems (Figure 4C). Among the six categories of KEGG pathways, metabolism contained the largest number of genes (975 genes), followed by environmental information processing (242 genes). It was found that it can be divided into 12 categories at the metabolism level, with carbohydrate metabolism (gene number: 219), global and overview maps (gene number: 171) and amino acid metabolism (gene number: 145) receiving the most gene function annotations. Carbohydrate metabolism mainly contained a total of 14 pathway information, including starch and sucrose metabolism (ko00500), glycolysis/gluconeogenesis (ko00010), pyruvate metabolism (ko00620), amino sugar and nucleotide sugar metabolism (ko00520), fructose and mannose metabolism (ko00051), pentose phosphate pathway (ko00030), galactose metabolism (ko00052), etc. In carbohydrate metabolism, starch and sucrose metabolism (ko00500), glycolysis/gluconeogenesis (ko00010), pyruvate metabolism (ko00620), and amino sugar and nucleotide sugar metabolism (ko00520) were dominant, and the number of related genes was 48, 45, 45 and 43, respectively. Present in ko00010, alcohol dehydrogenase [EC:1.1.1.1] (*adh*, including gene0084 and gene1515) and acetaldehyde dehydrogenase/alcohol dehydrogenase [EC:1.2.1.10 1.1.1.1] (*adhE*, gene3232) were involved in alcohol metabolism. In addition, there were 14 genes related to xenobiotics biodegradation and metabolism mainly containing a total of 10 pathway information. Among them, the ko00982 pathway was drug metabolism-cytochrome P450, and ko00980 was the metabolism of xenobiotics by cytochrome P450, both of which contained 2 related genes (alcohol dehydrogenase [EC:1.1.1.1], including gene0084 and gene1515) in the LTJ12 genome. At the environmental information processing level, there were 171 genes related to membrane transport, 70 genes related to signal transduction, and 1 gene related to signaling molecules and interaction.

### 3.5. Carbohydrate-active Enzyme (CAZyme) Annotation

Carbohydrate-active enzyme (CAZy) annotation was performed on the *L. pentosus* LTJ12 genome to identify the genes involved in alcohol metabolism. The results showed that 108 genes were identified from the CAZy family and distributed in 5 subfamilies. In *L. pentosus* LTJ12, glycoside hydrolases (GHs) had the most annotated genes, consisting of 43 genes. In addition, glycosyl transferases (GTs), carbohydrate esterases (CEs), auxiliary activities (AAs), and polysaccharide lyases (PLs) were also annotated, and the number of annotated genes were 32, 23, 9, and 1, respectively (Figure 4D).

Auxiliary activities (AAs) contained various oxidases and reductases. For example, enzymes with functions related to alcohol metabolism may belong to the AA3 family. In the genome of *L. pentosus* LTJ12, four AA3s were related with cellobiose dehydrogenase (EC 1.1.99.18), glucose 1-oxidase (EC 1.1.3.4), aryl alcohol oxidase (EC 1.1.3.7), alcohol oxidase (EC 1.1.3.13) and pyranose oxidase (EC 1.1.3.10). The four AA3 were gene0362, gene1012, gene1289 and gene2825, respectively. Additionally, one AA4 (gene0281) associated with vanillyl-alcohol oxidase (EC 1.1.3.38) was suggested to be potentially involved in catalyzing the oxidation of alcohols to aldehydes.

### 3.6. Alcohol-Related Genes of LTJ12 May Be Involved in the Degradation of Alcohol

To investigate the stress response and alcohol metabolism mechanism of LTJ12 at higher ethanol concentration (8% ethanol concentration), the expression of alcohol-related genes in LTJ12 was detected using ethanol-free culture conditions as a control. The results of quantitative real-time PCR (RT-qPCR) on the genes possibly related to alcohol metabolism were shown in Table 3 and Figure 5. Compared with ethanol-free culture conditions, the transcript levels of genes encoding alcohol dehydrogenase (gene 0155) was significantly up-regulated, suggesting that this gene might be involved in the degradation of ethanol by *L. pentosus* LTJ12.

### 3.7. LTJ12 Has Potential Probiotic Properties

The live microorganisms in probiotics have health benefits for both humans and animals [23]. In general, the requirements for probiotics are acid resistance through the gastrointestinal tract (reaching the gut), ability to survive in host gut or harsh manufacturing conditions, adherence in the gut, antibacterial activity against pathogens, use of carbohydrates, fats, and proteins, and health enhancements such as enhanced immunogenicity [24]. In addition to the probiotic properties of probiotics, food safety aspects should also be considered, and genomics could provide a new way to verify the absence of genes involved in the transfer of virulence or antibiotic resistance and the presence of genes involved in promoting health [25,26].

Bile salt tolerance and colonization ability are very important for the survival of probiotics in the host. A higher strain colonization capacity allows more strains to grow and multiply, and low survival rates in acid and bile salt environments mean that strains have difficulty surviving in the host. In our previous study, it was found that strain LTJ12 has good acid and bile salt resistance and adhesion ability, so we believe that it could have great potential for future development as probiotic strains [15]. To demonstrate the probiotic potential of *L. pentosus* LTJ12 with the ability to resist ethanol stress, we analyzed the probiotic genes related to stress resistance, immunomodulatory activity, etc. Through whole-genome sequencing analysis, *L. pentosus* LTJ12 has *dltA* (D-alanine-poly(phosphoribitol) ligase subunit 1 [EC:6.1.1.13]) and *dltD* (D-alanine transfer protein) genes, which are related to acid resistance [27]. In addition, *atpA* (F^−^ type H^+^-transporting ATPase subunit alpha [EC:3.6.3.14]), *cfa* (cyclopropane-fatty-acyl-phospholipid synthase [EC:2.1.1.79]) and *hisD* (histidinol dehydrogenase [EC:1.1.1.23]) were also found in the LTJ12 genome, and these genes were also associated with acid stress resistance [22].

Genes encoding bile salt hydrolase (choloylglycine hydrolase), such as gene0069, gene2334 and gene3057, were found in the genome of strain LTJ12. The presence of the bile salt hydrolase gene is an indicator of bile salt deconjugation ability. In the *Guidelines for the Evaluation of Probiotics in Food* issued by FAO/WHO, bile salt hydrolase activity is included among the desirable properties of probiotics, such as gastric acid resistance, bile acid resistance and adhesion. Removal of bile salts is listed as one of the properties guaranteed for safety [28]. From a microbial perspective, BSH (bile salt hydrolase) activity is important for secondary bile salt metabolism, which may promote bacterial bile resistance and favor host cholesterol metabolism by reducing host bile salt concentrations [29].

Factors that determine bacterial colonization of the host gut are surface hydrophobicity, excretion of secretases and lipopeptides, and utilization of epithelial cell-produced polysaccharides [30]. Bioinformatics analysis showed that LTJ12 encodes a variety of cell surface proteins, such as mucus-binding proteins (gene1500 and gene2813), lipoprotein signal peptidase (gene1625), elongation factor Tu (gene1920), etc [31]. Previous reports showed that *L. plantarum* and *L. pentosus* strains utilize elongation factor Tu (EF-Tu) and chaperonin GroEL to adhere on intestinal epithelial cells [12]. And the adhesion function of molecular chaperone DnaK in *Bifidobacterium* was confirmed [32]. In the whole genome of LTJ12, we found *tuf* (elongation factor Tu, gene1920), *groEL* (chaperonin GroEL, gene0675) and *dnaK* (molecular chaperone DnaK, gene1818), indicating the possibility of its adhesion. In addition, *groEL* (chaperonin GroEL, gene0675) and *groES* (co-chaperone GroES, gene0674) found in LTJ12 may also play important roles in the resistance to freezing and heat shock [33]. Also, we detected manganese/zinc transport system substrate-binding protein (*mntC*, including gene0997, gene2726 and gene3011), cell surface adherence protein, collagen-binding domain, LPXTG-motif cell wall anchor (gene2347 and gene2341), cell surface protein (gene2674). These genes appear to be involved in adhesion. Consistently, in our previous study [15], it was also demonstrated that LTJ12 had a certain role in adhesion.

The properties of probiotics are also related to exopolysaccharide (EPS) biosynthesis. EPS can facilitate niche adaptation as they promote auto-aggregation, attachment to abiotic or biotic surfaces and biofilm formation [34,35]. In addition, EPS also possesses various physiological functions, such as anti-inflammatory, antioxidant, antiviral and antiproliferative activities [36]. Finally, producing high concentrations of EPS alters the organoleptic properties of fermented products [37]. In our study, we found the presence of exopolysaccharide biosynthesis protein (gene1095, gene1096, gene1097, gene1116, gene1908 and gene1909) in the genome of strain LTJ12, explaining the possibility of LTJ12 having the above functions. Through our previous research, we found that LTJ12 can produce 101.90 mg/L exopolysaccharide when cultured at 37 °C for 24 h. Furthermore, we also found *dltB* (membrane protein involved in D-alanine export) and *dltD* (D-alanine transfer protein) genes, which are related to anti-inflammatory potential and immunomodulation [27].

The genome of LTJ12 was determined by PathogenFinder (https://cge.cbs.dtu.dk//services/PathogenFinder/, accessed on 1 June 2022) [20]. No matched pathogenic families were found in its genome, indicating that strain LTJ12 is not pathogenic. In our previous study, we also found that after being administered with LTJ12 at a dose of 5 × 10^9^ CFU/kg·d for 80 d, no death or poisoning was observed in the mice. CARD (Comprehensive Antibiotic Resistance Database, http://arpcard.Mcmaster.ca, Version 1.1.3, accessed on 1 June 2022), widely contains reference genes related to antibiotic resistance from various organisms, genomes, and plasmids, which can be used to guide the study on the antibiotic resistance mechanism of environment, human or animal flora. Through the CARD database annotation, it was found that the number of drug resistance genes annotated in the LTJ12 genome was 183, which can be mainly divided into 29 drug classes. Among them, the number of genes annotated to macrolide, fluoroquinolone, penam, tetracycline, peptide antibiotic, cephalosporin, acridine dye, lincosamide, nitroimidazole, cephamycin was the biggest, respectively 32, 17, 12, 10, 10, 9, 9, 8, 8 and 7. Due to the low stringency of the search criteria, most matches were not actually resistance genes. Since the CARD database mainly focuses on the resistance gene determinants of pathogenic bacteria, resistance genes of non-pathogenic bacteria such as those from *Lactiplantibacillus* are usually not included. Therefore, the CARD database has certain limitations for drug resistance gene search for non-pathogenic bacteria [28]. On the contrary, KEGG database search found 38 genes related to antimicrobial drug resistance on the LTJ12 chromosome, including *penP*, *vanX*, *vanY*, *abcA*, *abcA*, *penP*, indicating that it may have beta-lactam resistance, vancomycin resistance, cationic antimicrobial peptide (CAMP) resistance.

### 3.8. Comparative Genomic Analysis Suggests That LTJ12 Has More Functional Genes

Strain LTJ12 was compared to *L. plantarum* and *L. pentosus* on NCBI. Up to now, a total of 716 *L. plantarum* and 61 *L. pentosus* strains have genome assembly and annotation on NCBI (only eight of *L. pentosus* strains’ whole genome sequences with high assembly completion were available online publicly). Among them, the median total length of *L. plantarum* was 3.25 Mb, the median protein count was 2949, and the median GC content was 44.5%; while the median total length of *L. pentosus* was 3.74 Mb, the median protein count was 3266, and the median GC content was 46%. Combined with the strains in NCBI, it was found that the genomic information of strain LTJ12 was closer to *L. pentosus*. The general genomic characteristics of selected *L. plantarum* and *L. pentosus* for further comparative genomic analysis were described in Table 4.

Average amino acid identity (AAI) is an average based on the comparison of all orthologous amino acid sequences between two genomes, and is mainly used to assess the relationship between species at the genome-wide level. The larger the AAI value, the greater the similarity between the two samples. Generally, the AAI value between the same species is above 95%. AAI analysis was performed with strain LTJ12 based on the complete genomes of 8 strains of *L. plantarum* and 8 strains of *L. pentosus* (Figure 6). The results showed that the AAI values of strain LTJ12 and 8 strains of *L. plantarum* ranged from 88.46% to 88.79%, and the AAI value between strain LTJ12 and *L. plantarum* WCFS1 was the lowest (88.46%). However, the AAI values of strain LTJ12 and 8 strains of *L. pentosus* were all greater than 97.32%, which proved that strain LTJ12 had a higher homology with *L. pentosus*, and the AAI value between LTJ12 and *L. pentosus* SLC13 was the highest at 99.07%.

Pangenome is the general term for all genes of a species. The pan-genome represents the entire gene composition of a species and is the gene pool of all strains of the species. It is mainly composed of three parts: core genes, dispensable genes and unique genes. Pan-genomics analyzes bacterial genomes from a population perspective, studies the dynamic characteristics of bacterial genomes and analyzes the dynamic changes of bacterial genomes during evolution [38]. To study the genetic diversity of *L. pentosus*, in the present study, the pan-genome and core genes were analyzed. And the relationship between the number of core genes and pan-genes and the number of strains was investigated (Figure 7). In our study, we selected 6 strains of *L. pentosus* that can obtain gene nucleotides information on NCBI and compared them with LTJ12 for pan genomic analysis. The Venn diagram of the common and unique homologous genes of *L. pentosus* showed that the seven strains had 2235 common homologous genes, and each strain had 22–369 unique homologous genes. Among them, the LTJ12 strain had the largest number of unique homologous genes (gene number was 369), while *L. pentosus* SLC13 had the unique homologous genes with 120 (Figure 7A). The pan-genome size of the seven strains is 4746.0, and the core-genome size is 2235. As the number of strains increased, the number of pan-genes continued to increase (Figure 7B), while the number of core genes decreased and tended to be stable (Figure 7C). The results indicated that the *L. pentosus* population was an open genome. New genome sequencing will continue to increase the size of the total gene pool (Figure 7D). At the same time, it also reflects the huge number of pan-genomes of *L. pentosus* species. *L. pentosus* is widely distributed and exchanges various genetic materials with the outside world. The species has the diversity of genomic characteristics and the variability of evolutionary directions.

Based on the functional analysis of core genes, the common core genes were assigned to 31 COG types mainly associated with amino acid transport and metabolism, transcription, carbohydrate transport and metabolism, translation, ribosomal structure and biogenesis, cell wall/membrane/envelope biogenesis, replication, recombination and repair, inorganic ion transport and metabolism, etc. However, most genes were classified as function unknown. In particular, the LTJ12 strain in our study had more functional genes than other *L. pentosus* strains. The unique genes of the LTJ12 strain were assigned to 15 COG types, mainly associated with carbohydrate transport and metabolism, transcription, replication, recombination and repair, signal transduction mechanisms, defense mechanisms, cell wall/membrane/envelope biogenesis, etc. Also, the majority unique genes were categorized as function unknown.

Specifically, compared to other *L. pentosus*, only these genes exist in the LTJ12 strain isolated so far, such as *rpoN* (gene0733, encoding RNA polymerase sigma-54 factor), *msrA* (gene1795, encoding peptide-methionine (S)-S-oxide reductase [EC:1.8.4.11]), *hsdR* (gene0863, encoding type I restriction enzyme, R subunit [EC:3.1.21.3]), *hsdM* (gene0864, encoding type I restriction enzyme M protein [EC:2.1.1.72]), *hsdS* (gene0865, encoding type I restriction enzyme, S subunit [EC:3.1.21.3]), etc. Liu et al. [39] found that overexpression of *rpoN* encoding factor σ54 promoted *L. paraplantarum* L-ZS9 biofilm formation. σ54 has been reported to be directly or indirectly associated with biofilm formation in other bacteria and it may play a similar role in regulating biofilm formation by LTJ12. Yin et al. [40] found that *L. delbrueckii* subsp. *bulgaricus* incubation in MRS or milk also resulted in the production of different stress-responsive proteins. Among them, peptide methionine sulfoxide reductase (MsrA) was present in significantly larger amounts in *L. delbrueckii* subsp. *bulgaricus* after incubation in milk at 37 °C. Peptide methionine sulfoxide reductase (MsrA) is required to respond to oxidative stress. The Type I restriction-modification (R-M) systems consist of DNA endonuclease (HsdR, HsdM and HsdS subunits) and methyltransferase (HsdM and HsdS subunits). In certain Type I R-M systems, the *hsdS* sequences flanked by inverted repeats (called epigenetic inverts) undergo invertase-catalyzed inversions. Previous studies of *Streptococcus pneumoniae* have shown that *hsdS* inversions within clonal populations generate subpopulations that differ profoundly in methylome, cell physiology, and virulence [41]. In addition, gene 2754, the unique gene we found in strain LTJ12, which encodes alcohol dehydrogenase, may be related to alcohol metabolism. Similarly, gene3093 (gamma-D-glutamyl-meso-diaminopimelate peptidase) was also only found in LTJ12, which is associated with cell wall/membrane/envelope biogenesis. This gene was significantly up-regulated under ethanol stress (data not shown), indicating that it may enhance cell wall synthesis to resist ethanol damage, making it highly resistant to ethanol. In addition, genes related to defense mechanisms, including gene0863, gene0864, gene0865, gene0867 and gene3225, also only exist in LTJ12 and may also play a role in ethanol resistance. To elucidate the molecular mechanisms of the isolated strains as probiotics and improve their applicability, further functional studies of these genes will be carried out.

Collinearity generally means that certain regions between chromosomes in different species have similar gene arrangement. In the process of evolution, the factors that affect gene collinearity include chromosomal recombination and gene transposition. In general, the farther the evolutionary distance between species, the worse the gene collinearity, so the degree of collinearity between two species can be used as a measure of the evolutionary distance between them. At the same time, we can also obtain the structural variation of genomes between species during the evolution process through collinearity analysis.

The collinearity comparison analysis of the genomes of strain LTJ12 and 8 strains of *L. pentosus* downloaded from NCBI was performed with Mauve software. The Mauve alignment of the nine genomes identified approximately 17–22 Locally Collinear Blocks (LCBs, filled with different colors) separated by specific DNA stretches of different lengths (Figure 8). Identical colored regions are connected with lines that indicate which regions in these genomes are homologous. Mauve Alignment also detected a large number of chromosomal insertions, arrangements, and inversions. However, the synteny of genes was similar. Inversions and rearrangements are major evolutionary phenomena observed among *L. pentosus* strains and provide a complete picture of the genetic differences among strains colonizing different ecological niches [42].

*L. plantarum* WCSF1 was isolated from human saliva and first sequenced in 2001, and re-sequenced and re-annotated in 2012 [8]. *L. pentosus* SLC13, isolated from mustard pickles in Taiwan, is a high exopolysaccharide (EPS)-producing strain with broad-spectrum antimicrobial activity and the ability to grow under simulated gastrointestinal conditions [43]. Here, the whole genome sequences of *L. plantarum* WCSF1 strain and *L. pentosus* SLC13 were used as references to analyze the strain LTJ12 by Mauve software (Figure 9). Similarly, we found high homology was existed between *L. plantarum* WCSF1 and *L. pentosus* LTJ12. *L. pentosus* LTJ12 had good collinearity and similarity with *L. plantarum* WCSF1, and local insertions and deletions of *L. pentosus* LTJ12 occurred in locally collinear blocks.

Some genome insertions or deletions occurred between *L. pentosus* LTJ12, *L. pentosus* SLC13 and *L. plantarum* WCSF1. The insertion and inversion of short gene fragments suggest that gene recombination and transfer may have occurred in the strain during the evolution process [44], and the insertion or deletion of the genome may be related to the formation of an open genome [45]. The genomic differences between LTJ12 and the reference strain indicate that the genome has undergone recombination and transfer during evolution, but whether this phenomenon leads to the formation of an open genome in *L. pentosus* remains to be studied.

## 4. Conclusions

The similarity between *L. plantarum* and *L. pentosus* has raised many problems in research and application. The species *L. plantarum* and *L. pentosus* are genotypically closely related, and their phenotypes are always so similar that they are easily to be confused and mistaken. With the continuous development of sequencing technology and progress of bioinformatics, more and more *Lactiplantibacillus* genomes have been studied. In our study, we have distinguished *L. plantarum* and *L. pentosus* on the genome. By analyzing the genome of *L. pentosus* LTJ12 with the ability to resist ethanol stress, it was found that LTJ12 has multiple genes that may be related to alcohol metabolism, and these genes may be related to the high ethanol tolerance of *L. pentosus* LTJ12. In addition, their probiotic-related genes were predicted by genomic analysis, suggesting that it was potential probiotic candidates. Combined with comparative genomics analysis, strain LTJ12 has more functional genes than other *L. pentosus* strains mainly related to carbohydrate transport and metabolism, transcription, replication, recombination and repair, signal transduction mechanisms, defense mechanisms, cell wall/membrane/envelope biogenesis, etc. These unique functional genes, such as gene 2754 (encodes alcohol dehydrogenase, may be related to alcohol metabolism), gene3093 (encodes gamma-D-glutamyl-meso-diaminopimelate peptidase, associated with cell wall/membrane/envelope biogenesis) and other genes related to defense mechanisms, may enhance LTJ12 resistance to ethanol. This study provides a genetic basis for further studies on the mechanism of ethanol stress and the metabolic action of LTJ12, and thus shows promising prospects as a potential probiotic candidate.

## Figures and Tables

**Figure 1 foods-12-00035-f001:**
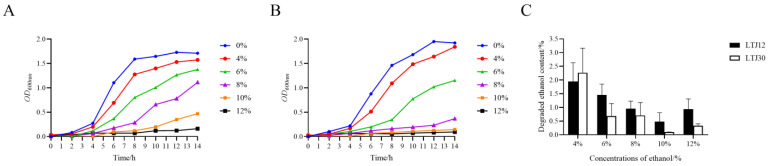
Effects of ethanol stress on strains. (**A**) Growth curve of LTJ12 during treatments of ethanol at different concentrations. (**B**) Growth curve of LTJ30 during treatments of ethanol at different concentrations. (**C**) Ethanol degradation ability of strains at different ethanol concentrations for 12 h. The values represent the means of the three replicates with the standard deviation (SD).

**Figure 2 foods-12-00035-f002:**
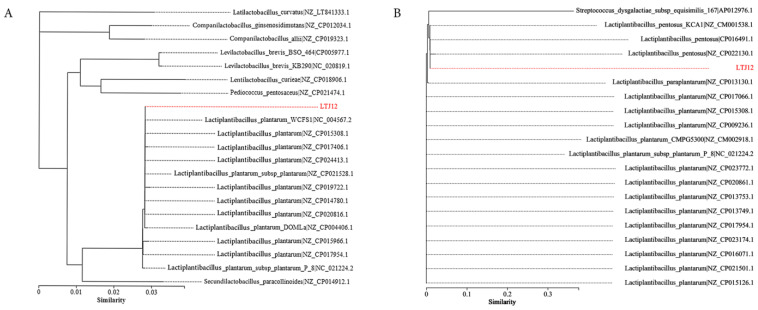
Phylogenetic tree of (**A**) 16S rRNA genes and (**B**) house-keeping genes. By comparing with the database, based on 16S rRNA sequence and housekeeping genes, 19 strains that were closest to LTJ12 at the species level were selected, and the NJ (Neighbor-Joining) method was selected using MEGA 6.0 software to construct the phylogenetic tree.

**Figure 3 foods-12-00035-f003:**
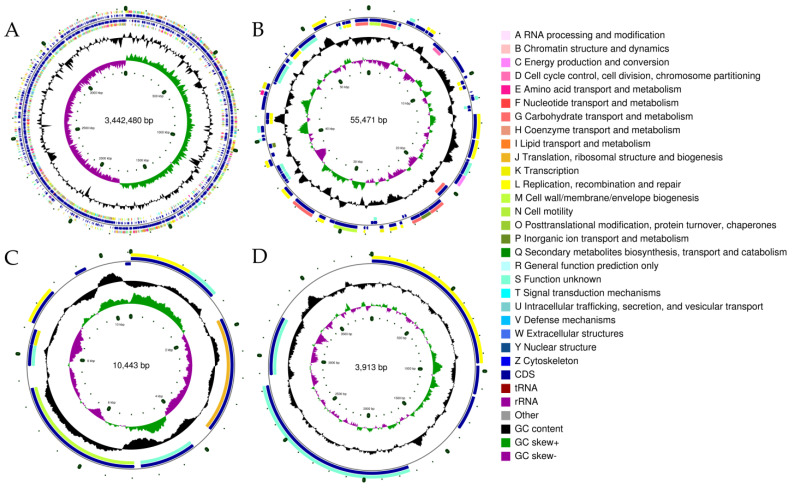
The whole genome of *L. pentosus* LTJ12. (**A**) Chromosome. (**B**) Plasmid 1. (**C**) Plasmid 2. (**D**) Plasmid 3. The genome map is composed of seven circles. From the outer circle to inner circle, each circle displays information regarding the genome of (1) forward COG function classification, (2) forward CDS, tRNA, rRNA, (3) reverse CDS, tRNA, rRNA, (4) reverse COG function classification, (5) G + C content, (6) GC skew and (7) the genome size identifier.

**Figure 4 foods-12-00035-f004:**
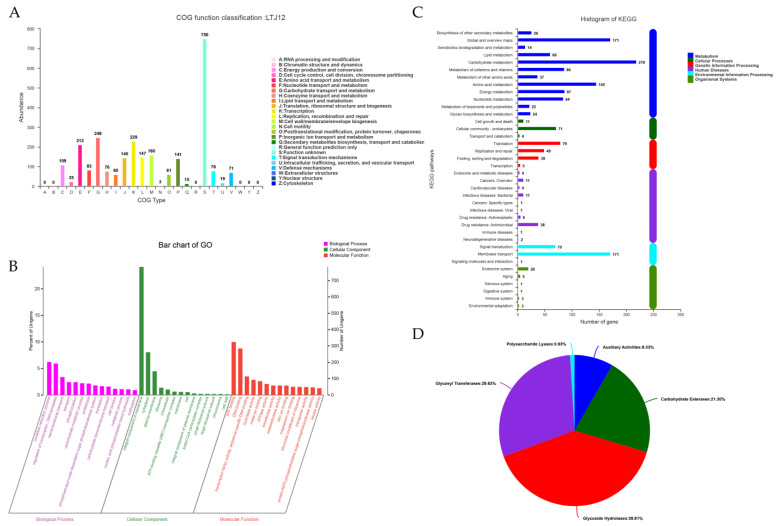
Functional annotation of *L. pentosus* LTJ12. (**A**) COG classification statistics histogram. (**B**) GO annotation classification statistics chart. (**C**) Pathway categorical statistics histogram. (**D**) Carbohydrate-activity enzyme annotation statistics chart.

**Figure 5 foods-12-00035-f005:**
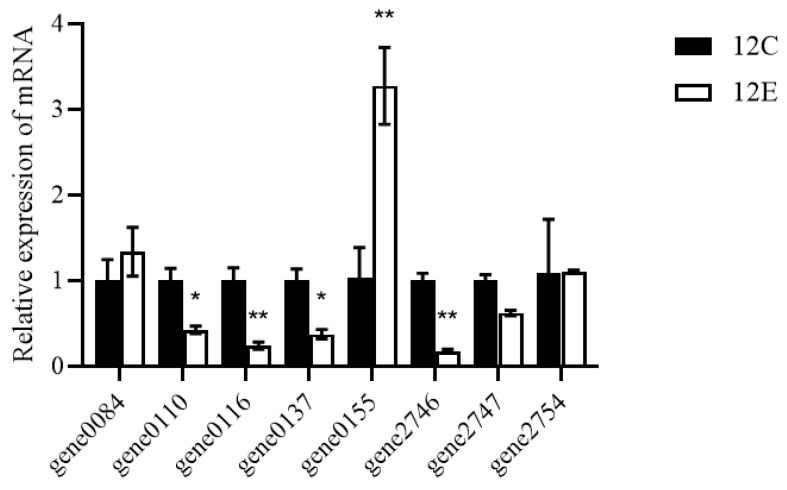
The transcript levels of alcohol metabolism related genes in *L. pentosus* LTJ12 cultured without or with ethanol. 12C stands for LTJ12 without ethanol, 12E stands for LTJ12 with 8% ethanol. The values represent the means of the three replicates with the standard deviation (SD). Asterisks represent significant differences from LTJ12 without ethanol. (statistical significance: * *p* < 0.05, ** *p* < 0.01).

**Figure 6 foods-12-00035-f006:**
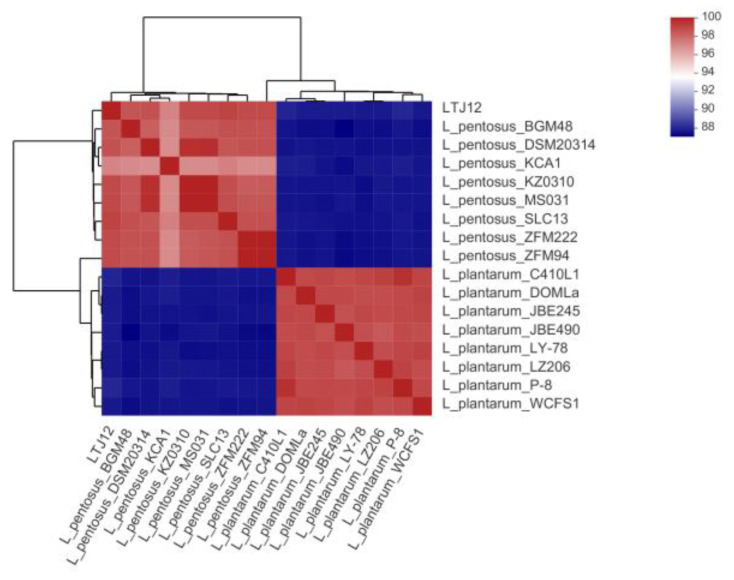
Heatmap showing the average amino acid identity (AAI) value between species of the strain LTJ12, 8 strains of *L. plantarum* and 8 strains of *L. pentosus*.

**Figure 7 foods-12-00035-f007:**
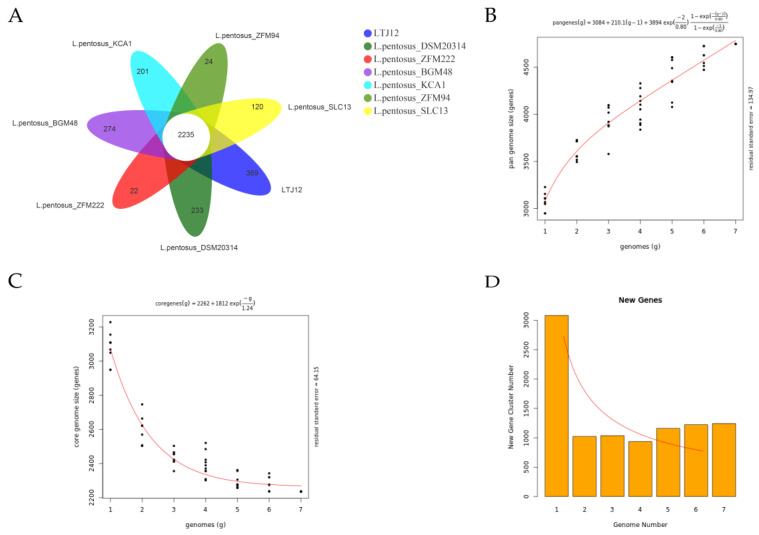
Pan and core genes of *L. pentosus*. (**A**) Venn diagram displaying the unique and core genes. (**B**) The curve of pan genome size changing with the number of genomes. The abscissa was the number of genomes, and the ordinate was the pan genome size. (**C**) The curve of core genome size changing with the number of genomes. The abscissa was the number of genomes, and the ordinate was the core genome size. (**D**) Column chart and curve of new gene size changing with the number of genomes. The abscissa was the number of genomes, and the ordinate was the new genome size.

**Figure 8 foods-12-00035-f008:**
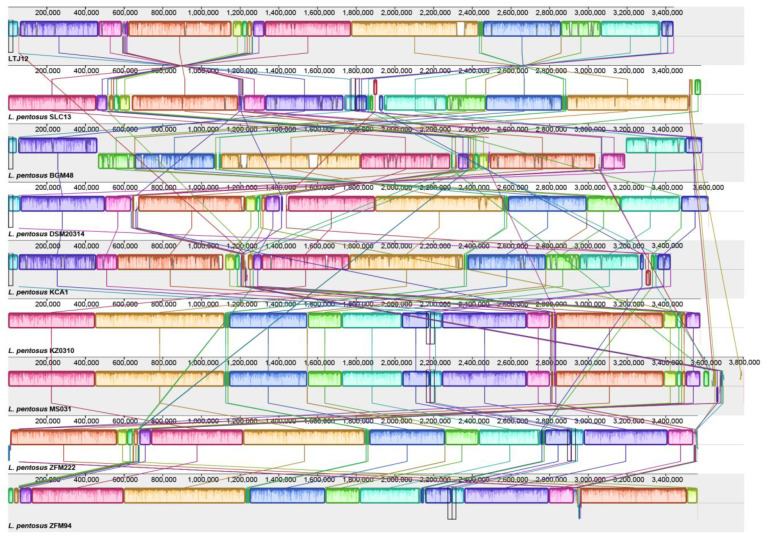
Mauve Alignment of all *L. pentosus* genomes available in the NCBI database.

**Figure 9 foods-12-00035-f009:**
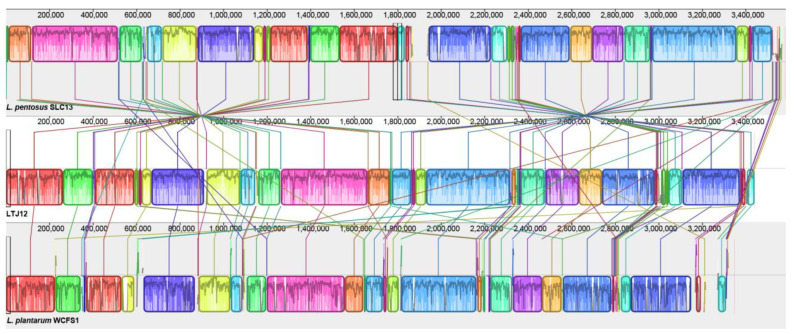
Mauve Alignment of the strain LTJ12, *L. pentosus* SLC13 and *L. plantarum* WCFS1.

**Table 1 foods-12-00035-t001:** The primers and sequences used in this study.

Gene ID	Gene Name	Forward Primer	Reveres Primer	Product Size
gene0084	*adh*	TACCTTCGCTGAGTTTATC	CTGAACCACCAGGAATAA	173
gene0110	-	ACGACGGTCTCACAACGC	AGGCCAAACGAGGCAAGT	175
gene0116	-	ACCATCCGACCAGAAGAA	GAGGGTTGCGACTGTGAG	113
gene0137	*gldA*	CACCAACCGCTGGCTTAT	GTCGCCTTCGCTTCAATA	181
gene0155	-	GTGAGACTTTTAGCCGTTTG	TTCTTCCGACTGACATCC	175
gene2746	*dhaT*	TATGACTAACGGTGACGAC	ACGAACTTGTAATGGGTG	156
gene2747	-	GTGACAACTGTTTGAAAGGGAT	CACCGTATGAGTCGTGAATG	155
gene2754	-	GTAATGATTCCAACGACAG	TTGACATCCAACAGAACTAA	134
-	16S rRNA	AAGGGTTTCGGCTCGTAAAA	TGCACTCAAGTTTCCCAGTT	247

**Table 2 foods-12-00035-t002:** Genome features of *L. pentosus* LTJ12.

Features	Genome
Genome Size (bp)	3,512,307
GC Content (%)	46.37
Gene (CDS) NO.	3248
Gene Len (bp)	2,835,894
Gene Average Len (bp)	873.12
Gene Density	0.92
GC Content in Gene Region (%)	47.59
Gene/Genome (%)	80.74
Intergenetic Region Len (bp)	676,413
GC Content in Intergenetic Region (%)	41.24
Intergenetic Len/Genome (%)	19.26
tRNA NO.	73
Type of tRNAs NO.	21
rRNAs NO.	16
16S rRNA	5
23S rRNA	5
5S rRNA	6
sRNAs NO.	38
NR annotation	3247
Swiss-Prot annotation	2168
Pfam annotation	2535
COG annotation	2593
GO annotation	2437
KEGG annotation	1541

**Table 3 foods-12-00035-t003:** Selected genes associated with alcohol metabolism.

Gene ID	Gene Name	Description	COG Description	GO Description	KO Description
gene0084	*adh*	oxidoreductase	alcohol dehydrogenase	oxidation-reduction process; oxidoreductase activity; zinc ion binding	alcohol dehydrogenase [EC:1.1.1.1]
gene0110	-	alcohol dehydrogenase	alcohol dehydrogenase	oxidation-reduction process; oxidoreductase activity; zinc ion binding	-
gene0116	-	oxidoreductase	alcohol dehydrogenase	oxidation-reduction process; oxidoreductase activity; zinc ion binding	-
gene0137	*gldA*	iron-containing alcohol dehydrogenase	Dehydrogenase	oxidation-reduction process; metal ion binding; glycerol dehydrogenase [NAD+] activity	glycerol dehydrogenase [EC:1.1.1.6]
gene0155	-	alcohol dehydrogenase	-	-	-
gene2746	*dhaT*	1,3-propanediol dehydrogenase	alcohol dehydrogenase	oxidation-reduction process; metal ion binding; 1,3-propanediol dehydrogenase activity	1,3-propanediol dehydrogenase [EC:1.1.1.202]
gene2747	-	aryl-alcohol dehydrogenase	alcohol dehydrogenase	oxidation-reduction process; zinc ion binding; aryl-alcohol dehydrogenase (NAD+) activity	aryl-alcohol dehydrogenase [EC:1.1.1.90]
gene2754	-	alcohol dehydrogenase	alcohol dehydrogenase	oxidation-reduction process; metal ion binding; lactaldehyde reductase activity	-

**Table 4 foods-12-00035-t004:** Genome features of selected *L. plantarum* and *L. pentosus*.

	Strains	Accession	Size (Mb)	CDS	GC%
*L. plantarum*	*L. plantarum* WCFS1	NC_004567.2	3.35	3041	44.45
	*L. plantarum* strain LY-78	NZ_CP015308.1	3.13	2830	44.77
	*L. plantarum* strain JBE245	NZ_CP014780.1	3.26	2919	44.5
	*L. plantarum* DOMLa	NZ_CP004406.1	3.21	2897	44.67
	*L. plantarum* strain LZ206	NZ_CP015966.1	3.26	2942	44.5
	*L. plantarum* strain C410L1	NZ_CP017954.1	3.39	2996	44.42
	*L. plantarum* subsp. *plantarum* P-8	NC_021224.2	3.25	2951	44.55
	*L. plantarum* strain JBE490	NZ_CP020861.1	3.20	2820	44.57
*L. pentosus*	*L. pentosus* strain SLC13	NZ_CP022130.1	3.58	3084	46.41
	*L. pentosus* strain BGM48	NZ_CP016491.1	3.68	3236	46.13
	*L. pentosus* KCA1	NZ_CM001538.1	3.43	5931	46.40
	*L. pentosus* strain DSM 20314	NZ_CP032757.1	3.67	3199	46.33
	*L. pentosus* strain ZFM222	NZ_CP032654.1	3.70	3198	46.14
	*L. pentosus* strain ZFM94	NZ_CP032659.1	3.69	3179	46.19
	*L. pentosus* strain MS031	NZ_CP043671.1	3.81	3303	46.10
	*L. pentosus* strain KZ0310	NZ_CP044245.1	3.87	3372	46.03

## Data Availability

The data presented in this study are available within the article.

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
