# Peer review of "Comparative Genomics Analysis Provides New Insights into High Ethanol Tolerance of Lactiplantibacillus pentosus LTJ12, a Novel Strain Isolated from Chinese Baijiu"

_foods, 2022, doi:10.3390/foods12010035_

Round 1
Reviewer 1 Report
This manuscript describes the genomic characterisation of a Lactiplantibacillus pentosus strain. Generally the work is good, but there are some problems with the data that must be addressed. The quality of the English is OK, but some sections are difficult to understand, or confusing. I have not addressed all of the language problems, but list some of them as examples below.
My key concerns are highlighted in bold. The work as it is requires a significant rewrite.
L24: the number of predicted coding genes
L26: and an assessment of metabolic pathways was performed
L29: LTJ12 is L. pentosus and not L. plantarum,
L32: wall/membrane/envelope biogenesis. (delete the ", etc")
L35: "with special excellent alcohol" needs to be changed
L37: change "tolerance and alcohol metabolism, the findings in this study will" to "tolerance and alcohol metabolism. The findings in this study will..."
No mention of the presence of the antibiotic resistance genes in the abstract.
L45: I don't know what "the primary bacteria" means
L46: "Lactobacillus are fastidious Gram-positive bacterium" to "Lactobacilli are fastidious Gram-positive bacteria"
L55: and the rest of this paragraph, check spelling of Lactiplantibacillus. After the first mention of the species can abbreviate to L.
Examples of difficult to understand sentences include
"so it is of great significance to conduct rigorous identification" or
"With the rapid development of science and technology, the exploration and study of LAB has also begun to explore from the basic research of some physiological and biochemical experiments to the field of related mechanism research, involving the research of genomics, transcriptomics, and metabolomics, etc"
L72: This isn't the definition of comparative genomics, but instead an indication of why it is useful. More accurate version here: https://www.genome.gov/about-genomics/fact-sheets/Comparative-Genomics-Fact-Sheet
L74-90: could be one short sentence after the one about the problems around the use of 16S to distinguish lactiplantibacilli
L103: The good LAB required in the production and fermentation should have good fermentation performance and strong tolerance ?
L108: "screened from" should be isolated from, used throughout.
L115: the databases were not annotated by PacBio and Illumina data. The assemblies from the PB and Illumina data were then annotated using the databases.
L123: "in the early stage"?
L129: and throughout, mention of alcohol, should this be ethanol?
One part says 12 hours, the other 14 hours?
L132: how was the alcohol analysis performed?
L148-159: This section reads as if the 400 bp fragmented DNA was used for Illumina and PacBio sequencing. What version of PB chemistry and movie length used?
L163: Need to explain that this is the PB data, otherwise reads like this is the pipeline for the Illumina data.
L167: I doubt HGAP and canu were both used?
L167: How was the circularisation performed?
No information in any of the materials and methods on settings used for the analysis. No accession number for the sequencing data. No one can replicate the work described.
L182: No explanation of why these 8 strains were selected? I think respectively should read 8 of each of L. plantarum and L. pentosus?
L184: no details on how the AAI comparisons were performed.
L197: respectively doesn't make sense here and alcohol mentioned again but not what type.
L202:does 16S expression stay constant to use as the endogenous reference gene?
L207: screened/isolated
L210: "Based on previous studies, here,"?
Figure 1 would benefit from colour, and reads as if LTJ30 does a better job of breaking down alcohol than the LTJ12 in Fig 1C?
L227: Delete afterwards
L228: Delete however
Fig 2: why are these strains in the tree? Difficult to read, font needs to be increased. No detailed information on how the tree was generated, or why these housekeeping genes were selected, Different strains used in A and B, including pediococcus and streptococcus?
L240: "The genome contributes" changed to Genome analysis contributes to a clearer understanding of.....
L248: sRNAs mentioned here and in Table 1, but no info on how they were detected?
L249, would you expect 16 rRNA operons in these species?
L265: "The number of LTJ12 coding genes in COG category was 4," Is this number correct? Doesn't match the subsequent numbers.
L279: were predicted to encode instead of the encoding genes. (none of the functions were proven)
L282: "According to the comparison results"?
L300: a word is missing after "most"
L300: In molecular functional?
L327:" In addition, there were 14 genes related to xenobiotics biodegradation and metabolism mainly containing a total of 10 pathway information"?
L329: "Among them" what is them in this case?
Figure 4: Information needs to be made larger, difficult to read in the publication format
Table 2: No information on why this subset of genes were selected for the qPCR. No information on the primer designs, size of PCR products, cycling conditions used etc.
Figure 5: What do the error bars represent?
L372: The probiotics are the live microorganisms?
L384: The definition of probiotics has to do with their positive health impacts. The survival is a different characteristic of the strains.
L396: delete "by searching"
L402: "From a microbial perspective, BSH (bile salt hydrolase) activity is important for secondary bile salt metabolism, which may promote bacterial bile resistance and favor host cholesterol metabolism by reducing host bile salt 404 concentrations, but excess deconjugated bile salts may interferes with host digestive activity or gut health and should further quantitatively study the production of deconjugated bile salts by BSH-positive strains" ?
L452: Discussion on the detection of the antibiotic resistance genes, but no mention of any legal problems to do with the strain due to this, or no tests of MICs.
L464: mentions 61 pentosus genomes but then only 8 in the next sentence?
L521: No mention of checking those genes against the other 61 pentosus genomes
Fig 7: This is a problem. No mention of B/C/D in the text. And of the 8 strains mentioned only 6 included int he pangenome comparison with the LTJ12. Strangely, the 2 omitted (KZ0310 and MS031) showed very high synteny with LTJ12 in Fig 8. Fig. 8 shows the importance of starting the genomes at the same point for the Mauve alignment. This is the kind of detail missing from the information on circularisation of the genomes and the informatics analysis.
This is especially important as a key conclusion of the paper is LTJ12 has more functional genes than other L. pentosus strains.
I don't see the purpose of including Fig 9 and the associated text?
L595: "The similarity between L. plantarum and L. pentosus has given people a lot of problems in research and application." This doesn't provide me with any information on what the problems are.
Reviewer 2 Report
The content of the presented work does not match the theme and is too extensive. According to the subject of the work, the sequencing of the genome of the new strain was to be used to indicate resistance to alcohol stress and the ability to degrade alcohol. However, too immense description of the entire genome of the tested strain was presented in parallel, along with a comparison of the strain with others LAB available in databases. As a consequence, proved that the previous species identification was not correct. The text presents too much detailed information on the characteristics of the genome, a whole range of genes, the presence of which suggests that the tested strain has several probiotic properties. Too much information makes the article hard to read. I believe that a great deal of work has been done, especially in bioinformatics, which can be used to create even more than one article. I suppose that if the authors want to stay on the selected topic, they should significantly shorten/modify the manuscript, converting a large part of the results to supplementary files.
I suggest that the manuscript should be re-checked for correct English. There are grammatical errors as well as syntax errors. Typos, especially conspicuous in the genus name - lines 55 - 61. The expression "etc" often repeated in the text does not mean anything specific, is too general and therefore redundant in scientific descrition.
The description of M&M section is not accurate enough. In particular, there is no information about the replicates performed (in subsections 2.2 and 2.7). There is information on this subject in the caption of Fig. 5 and no information about the standard deviation in the caption of Fig. 1.
Round 2
Reviewer 1 Report
I do not think the authors have addressed many of the issues highlighted in my original review. I include some examples below. At a certain point I stopped checking the responses as it appeared many had not been addressed.
Response 4: As suggested by the reviewer, we have supplemented this part in the materials and methods in the manuscript. Searching for PRJNA907194, PRJNA, or 907194 on NCBI didn’t get me a hit for this project.
Response 13: “This number is correct. It can match the subsequent numbers.”. How can you have 4 codign genes in COPG category but 2593 genes in the genome?
Response 17: “Figure 7 B/C/D has been supplemented in the text.”.
- Has it? The only mention was to add to the sentence on L545?
“Pan genome analysis is based on the gene nucleotides of each strain on the online tool of Majorbio Cloud Platform. There is no nucleotide faa file of KZ0310 and MS031 in NCBI, but only a fasta file of genome sequence. So the 8 strains mentioned only 6 included in the pangenome comparison with the LTJ12 but all 8 strains performed collinearity analysis. And we explained that LTJ12 has more functional genes than other L. pentosus strains in detail in the manuscript.”
- I think this means the authors are saying there is no annotated genome for the 2 strains? These two strains (KZ0310 and MS031) have the biggest genome sizes of the 8 pentosus genomes (Figure 4) being compared to LTJ12. Not including them in the pangenome analysis, but showing a high level of synteny according to the Mauve alignments, and then claiming more functional genes for LTJ12 doesn’t make sense to me.
Response 20: Some of the suggested improvements have been made to the manuscript, others have not. I haven’t checked most of them, but these are examples below of changes that the authors suggest have been addressed, but as far as I can tell have not been. This lack of attention to detail/transparency is frustrating as the manuscript took some time to review.
- This includes comments made by both me and Reviewer 2 around typos and syntax (e.g., Lactiplanatibaillus).
- Another example is my comment on “L115: the databases were not annotated by PacBio and Illumina data. The assemblies from the PB and Illumina data were then annotated using the databases.”. A section has been deleted, but now the sentence does not make sense.
- “L167: I doubt HGAP and canu were both used?” this was not addressed. I'm not saying this is wrong, but is not clear.
- L148-159: This section reads as if the 400 bp fragmented DNA was used for Illumina and PacBio sequencing. What version of PB chemistry and movie length used?”. This doesn’t sem to have been addressed either?
Reviewer 2 Report
Response to the authors comments:
Comment to point 1: I believe that, in accordance with the topics, the results from the sequencing performed should be concerned with potential genetic markers that allow differentiation of the genus Lactoplantibacillus, but especially about potential genes responsible for alcohol tolerance and metabolism. In my opinion, the content of the manuscript goes well beyond this scope. However, this is a purely subjective statement, irrelevant to the fact that Foods has no restrictions on the length of manuscripts.
Comment to point 2: I agree that the genus name is used correctly already in the title. I suppose that typos have crept into its name in several places: Lactiplanatibaillus (lines: 58, 59, 62, 63). I was not able to find genus Lactiplanatibaillus on the webpage: http://lactobacillus.ualberta.ca/
